# *The Stochastic Parrot on LLM's Shoulder:*
# A Summative Assessment of Physical Concept Understanding

## Abstract

In a systematic way, we investigate a widely asked question: *Do LLMs really understand what they say?*, which relates to the more familiar term *Stochastic Parrot*. To this end, we propose a summative assessment over a carefully designed physical concept understanding task, Physico. Our task alleviates the memorization issue via the usage of grid-format inputs that abstractly describe physical phenomena. The grids represents varying levels of understanding, from the core phenomenon, application examples to analogies to other abstract patterns in the grid world. A comprehensive study on our task demonstrates that: (1) state-of-the-art LLMs lag behind humans by ∼40%; (2) the stochastic parrot phenomenon is present in LLMs, as they fail on our grid task but can describe and recognize the same concepts well in natural language; (3) our task challenges the LLMs due to intrinsic difficulties rather than the unfamiliar grid format, as in-context learning and fine-tuning on same formatted data added little to their performance. Our data is released (see Supplementary Material in the submission) for public research.

## 1 Introduction

Recent years have witnessed remarkable advancements in large language models (LLMs) (Brown et al., 2020; Achiam et al., 2023; Team et al., 2023). Thanks to the substantial model capacity and massive training data, LLMs have achieved new state-of-the-arts on a variety of NLP tasks, even surpassing humans on some of them (Min et al., 2023; Chang et al., 2024). Nowadays the application of LLMs has become widespread, facilitating daily work and life, and profoundly influencing people's work and lifestyles (Bommasani et al., 2021; Peng et al., 2024; Demszky et al., 2023).

On the other hand, despite the great success of LLMs, many researchers argue that *LLMs may not really understand what they claim they do* (Bender & Koller, 2020; Bender et al., 2021; Bommasani et al., 2021; Mitchell & Krakauer, 2023) due to their strong memorization ability. In particular, Bender et al. (2021) questioned whether LLMs are just *Stochastic Parrots* that repeat words based on correlations without true understanding. This argument has been acknowledged by many research papers and dozens of them even include this term in their titles.[1] Unfortunately, to our best knowledge, there are no quantitative experiments to verify the stochastic parrot phenomenon in LLMs. Existing studies indicate that LLMs may fail on one particular challenging task (Chakrabarty et al., 2022; Shapira et al., 2023; Hessel et al., 2023; Tong et al., 2024), but they do not demonstrate that LLMs claimed to understand those tasks by providing a controlled and paired evidence.

This paper aims to provide quantitative evidence to validate the argument of stochastic parrot in LLMs. To this end, from the perspective of educational and cognitive psychology, we first employ the approach of summative assessment (Black & Wiliam, 1998a;b) to measure understanding in LLMs. Its key idea is to design various tasks that test different understanding levels regarding a specific concept. Following the principle of Bloom's taxonomy (Armstrong, 2010; Krathwohl, 2002), we design tasks that reflect different levels of understanding. Consequently, we develop Physico, a task designed to assess understanding of basic physical concepts from high school such as *Gravity*. Our focus on physical concepts stems from both their fundamental relevance to important topics of

---

[1] https://scholar.google.com/scholar?hl=en&q=llms+are+stochastic+parrot.

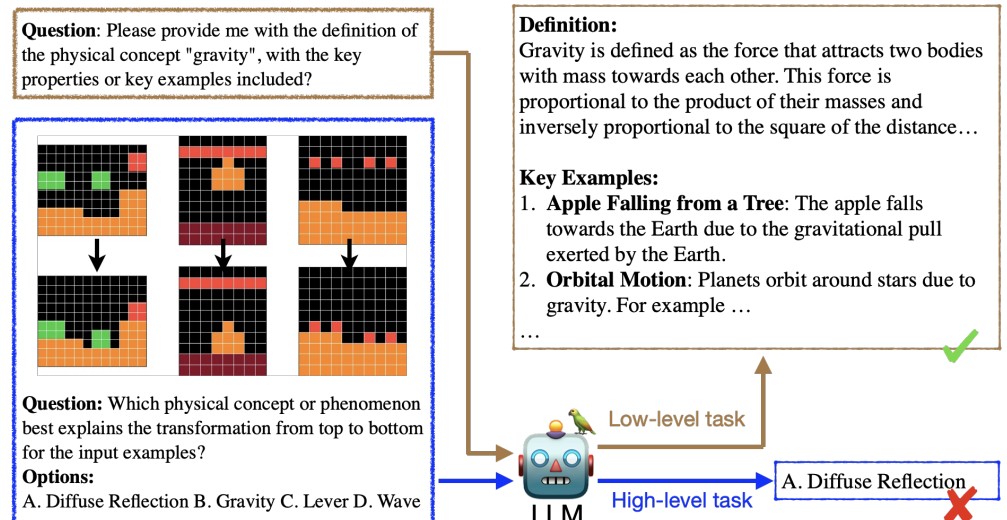

Figure 1: Illustration of a "Stochastic Parrot" by our PHYSICO task consisting of both low-level and high-level subtasks in parallel. For a concept *Gravity*, an LLM can generate its accurate description in natural language, but cannot interpret its grid-format illustration.

world models and embodied systems (Savva et al., 2019; Duan et al., 2022; Xiang et al., 2023), and their rich denotations and connotations that enable effective design of summative assessment tasks.

Specifically, PHYSICO includes two subtasks corresponding to two coarse levels of understanding in Bloom's taxonomy, as shown in Figure 1. One is the low-level understanding subtask in the natural language format, aimed at measuring the remembering (or memorization) ability of LLMs. The other involves the same concepts but in an abstract representation format inspired by (Chollet, 2019), which is designed to measure the high-level understanding beyond remembering of LLMs.

We conduct comprehensive experiments on PHYSICO with representative open-source and commercial LLMs.[2] We obtain two key findings: 1) SOTA LLMs perform perfectly on the low-level understanding subtask (>95% in Accuracy) but lags behind humans by a large margin (∼40% in Accuracy) on the high-level subtask, which verifies the stochastic parrot phenomenon in LLMs. 2) Further analysis shows that our high-level subtask challenges LLMs due to the intrinsic difficulty of deep understanding rather than the unfamiliar format.

This paper makes the following contributions:

- We introduce a psychology-appealing approach (summative assessment) and a corresponding task PHYSICO to measure the understanding of LLMs.
- Based on PHYSICO, we provide a quantitative experiment to successfully verify the stochastic parrot phenomenon in LLMs.
- As a by-product, our work presents a challenging comprehension task for existing text-only and multimodal LLMs, which establishes a substantial performance gap between humans and machines.

## 2 MEASURING CONCEPT UNDERSTANDING VIA SUMMATIVE ASSESSMENT

It is intrinsically challenging to measure the extent to which LLMs understand a sentence or concept. Indeed, Bender & Koller (2020) provide a definition of "understanding" from a linguistic perspective, but this definition depends on another abstract and unmeasurable term, "*meaning*". Therefore, even with this definition, accurately measuring "understanding" remains elusive.

We approach the measurement of whether LLMs understand a concept from an educational and cognitive perspective, using **summative assessment** (Black & Wiliam, 1998a;b; Harlen & James, 1997). Summative assessment is widely used by educators as an appealing strategy to evaluate students' understanding and knowledge acquisition in educational and cognitive psychology. For

---

[2]Throughout this paper, LLM refers to either standard text-only LLMs or large multimodal models for simplicity.

example, when middle school physics teachers want to know whether a student truly understands the concept "*Gravity*", they would design a series of questions specifically related to the concept of gravity to assess comprehension, *e.g.*, the properties like inverse square law and examples like orbital motions. If a student struggles to answer many of these questions, the teacher may conclude that the student does not understand the concept well or has a poor grasp of it.

We extend the idea of summative assessment to evaluating the concept understanding from humans to machines. Formally, assume $\mathcal{S}$ denotes an intelligent system and $\mathcal{C}$ is a specific concept. To evaluate the extent how $\mathcal{S}$ understands the concept $\mathcal{C}$, our summative assessment includes the following two steps:

- *Task design towards* $\mathcal{C}$: design several concept understanding tasks, each of which consists of several questions manually created towards understanding the concept $\mathcal{C}$.
- *Evaluating* $\mathcal{S}$: ask $\mathcal{S}$ to answer the questions from the tasks and calculate its accuracy.

**Requirements for Validity** The success (validity) of the proposed evaluation approach highly depends on the task design (Black & Wiliam, 1998a;b). For example, if the questions are too easy, even a weak system could answer them correctly. This leads to an overestimation of the system's understanding capabilities, making the assessment ineffective. To ensure good validity, we adhere to the principles outlined in summative assessment (Black & Wiliam, 1998a;b) for task design:

- *Alignment with evaluating objectives*: the questions should be related to the targeted concept, and should measure the specific knowledge about the targeted concept.
- *Different difficulty levels*: the questions should be with different difficulty levels from easy to difficult level, to ensure that the evaluation results have distinctiveness for different systems.
- *Variety*: the questions should reflect various understanding aspects of the targeted concept; addressing both its denotation and connotation.
- *Simplicity*: while not mandatory, a simpler benchmark for humans can more effectively highlight the issue faced by current models, i.e., the stochastic parrot effect in LLMs.

## 3 Task Design and Dataset Construction

### 3.1 Task Design Principle

We borrow the idea of Bloom's taxonomy (Krathwohl, 2002; Armstrong, 2010) from education research to fulfill the requirements for task design in Section 2, so as to ensure the assessment validity. Bloom's taxonomy offers an ideal principle to these requirements with an ordering of six cognitive skills (from low to high level) for knowledge understanding: *Remembering, Understanding, Applying, Analyzing, Evaluating and Creating*.

Generally, it is nontrivial to strictly follow this principle since there is no clear boundary among the last four skills of understanding. As a result, we group the last four high-level skills into one and consider the following two levels of understanding:

- *Low-level Understanding*: covering the two lowest-level skills in Bloom's taxonomy, *i.e.*, retrieving relevant knowledge from long-term memory and rephrasing in one's own words.
- *High-level Understanding*: covering the aspects for understanding the knowledge beyond memorization. As shown by the examples in Section 3.2.2, our tasks directly correspond to a spectrum from the understanding level of applying to the level of analyzing in Bloom's taxonomy, *e.g.*, *applying* the knowledge to explain a physical phenomenon, *analyzing* a concrete property of a concept in a generalized and abstract manner,[3].

Based on these two levels, we design the following PHYSICO task for summative assessment.

### 3.2 Our PhysiCo Task

PHYSICO is essentially a physical concept understanding task, which primarily targets on 52 physical concepts or phenomena: *e.g., gravity, light reflection, acceleration, buoyancy, inertia, etc* (see Appendix A for the full list). Our focus on physical concepts is motivated by two main reasons: 1)

---

[3]For example, the flow of electric current can be abstracted as *moving* from high potential to low potential.

understanding physical concepts is critical for intelligent systems to interact with the world, which is ultimate goal of embodied AI (Savva et al., 2019; Duan et al., 2022; Xiang et al., 2023); 2) designing tasks centered around physical concepts allows us to more easily control different levels of understanding and ensure the diversity of each concept.

For each physical concept, PHYSICO involves both low-level understanding subtasks and high-level subtasks, following our task design principles.

### 3.2.1 LOW-LEVEL UNDERSTANDING SUBTASKS

**Physical Concept Selection (text)**    First, to evaluate whether an LLM possesses the knowledge of our included concepts, we design a task to recognize a concept from its corresponding Wikipedia definition. Specifically, we manually masked the synonyms of the concept with placeholder [PHENOMENON]. Meanwhile, highly relevant entities were masked as [MASK] to alleviate shortcuts. For example, in the definition of *Gravity*, the terms "gravity" and "gravitation" were masked as [PHENOMENON], while "Isaac Newton" was masked as [MASK]. Details can be found in Appendix B. We then present the LLMs with the same four choices as in our following high-level subtasks.

**Physical Concept Selection (visual)**    Second, we evaluate if the LLMs can recognize our concepts represented with real-life pictures. To this end, we query our concepts on Google image search, and select the images that reflect the same core properties and examples annotated in our following high-level tasks. This results in 100 examples. We construct the same four-choice instances as above.

**Physical Concept Generation**    Finally, we directly ask the LLMs to generate the description of a concept with its core properties and representative examples. For instance, the concept *Gravity* is described as "*a force that pulls objects with mass towards each other*", followed by the example "*an apple falls to the ground*" as shown in Figure 1. We then evaluate the performance of LLMs by measuring the quality of the description and its coverage of knowledge required by our PHYSICO and we employ both automatic and human metrics as presented in Section 5.2. This provides a quantitative measure of the knowledge LLMs can recall in the context of our assessment.

### 3.2.2 HIGH-LEVEL UNDERSTANDING SUBTASKS

The low-level subtasks are depicted in natural language thus are likely to be remembered by the LLMs due to their extensive training data. To assess whether the LLMs possess a deep understanding of the knowledge, we require the subtasks that can 1) represent the high-level understanding skills; 2) avoid the effects of memorization.

The Abstraction and Reasoning Corpus (ARC) (Chollet, 2019) provides a compelling way by using grids (or matrices) instead of texts to represent a concept. While the LLMs have seen matrices during pre-training, the data is less likely to be correlated to physical concepts. We hence adopt this idea to represent our subtask as abstract representations in the grid world that associate to the key properties of a physical concept.

**The PHYSICO-CORE Set**    Our first subtask aims to cover the core properties or most representative examples/applications of the assessed concepts. To ensure our set remains generally comprehensible to humans, we maintain a high school-level difficulty and selected 52 common physical concepts within the curriculum. To enhance the diversity and richness, five annotators have labeled multiple core aspects of each concept. For example, the annotated core aspects of *Gravity* include *attraction between two bodies, motion on an inclined plane, objects falling to grounds* and *orbital motions*.

For each aspect of a concept, the annotator is asked to draw several pairs of abstract grid representations. The aspect of the concept is guaranteed to be illustrated by the pair, such that it explains the transformation from the input to the output. For example, Figure 1 forms a direct abstract visualization of the *Gravity* concept from textbooks, *i.e.*, *apple falling from a tree*. This results in 1,200 paired grid examples for the 52 concepts, which form 400 3-shot instances.

Figure 2 presents two examples from this subtask that delve deeper into the concept of *Gravity* compared to Figure 1. The top example demonstrates an application of the *inverse square law of gravity*. The bottom one presents a parabola, linking the knowledge of *gravity* to *inertia*. These

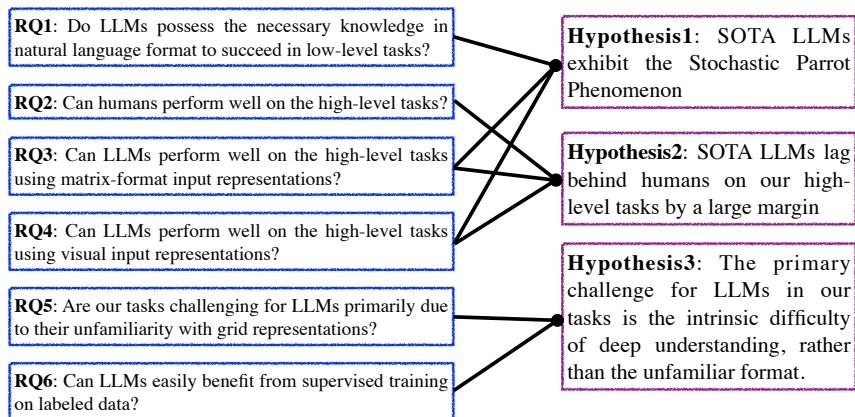

Figure 3: Overview of the research questions answered in our study and their relationships.

examples demonstrate the difficulty of inferring their ground-truth labels solely by recalling the concept of *Gravity* without high-level understanding skills.

**The PHYSICO-ASSOCIATIVE Set**   Many instances in the original ARC dataset can be solved via association or analogy to physical concepts. Therefore, as a second source of subtasks, we ask annotators to manually pick input-output grids from ARC that can evoke their associations to specific physical concepts and assign these concepts as ground-truth labels. Different from PHYSICO-CORE, we adopt an open-coding schema and allow the inclusion of new concepts during annotation. The annotators have reviewed 500 ARC instances to filter out the required ones. After cross-validation to ensure agreement, it results in a collection of 200 instances with physical concept labels.

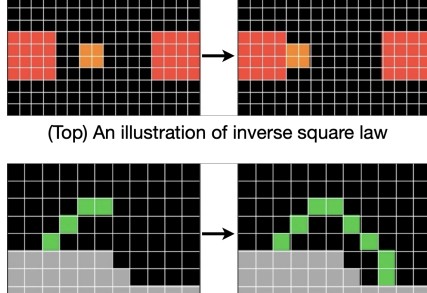

(Top) An illustration of inverse square law

(Bottom) An illustration of parabolas

Figure 2: Examples of input-output grids labeled as *Gravity*, with increasing difficulty levels.

This relabelling approach covers additional 15 physical concepts. The resulted subtasks have each example represent an abstract aspect of a concept with possible distracting information. Consequently, the resulted task is more subjective hence more challenging than the PHYSICO-CORESet.

**Creation of Classification Tasks**   We create *four-choice* tasks on the annotated data. Each instance consists of 3 unique grid pairs as input examples. This results in 200 instances for PHYSICO-CORE development set, 200 instances for PHYSICO-CORE test set, and 200 instances for ASSOCIATIVE respectively. For each instance, we select three additional labels from our concept pool, along with the ground-truth label, as candidate options. We manually avoid ambiguity during the negative sampling. For example, if *Gravity* is the ground-truth, concepts like *Magnet* will not be sampled.

## 4 OVERVIEW OF OUR STUDIES

In the following sections, we conduct a series of studies on our PHYSICO tasks. Our studies are organized into six *Research Questions (RQs)*, through which we aim to answer three *Hypotheses (Hs)* as shown in Figure 3. In summary, we propose to:

(1) Examine the quantitative disparity in LLMs' performances between low-level (RQ 1) and high-level subtasks (RQ 3, RQ 4). This aims to highlight **the existence of stochastic parrot phenomenon** in LLMs' understanding of physical concepts.

(2) Assess the performance gap between LLMs (RQ 3, RQ 4) and humans on our high-level subtasks (RQ 2). This aims to demonstrate that LLMs **fall significantly short of human understanding**.

(3) Investigate the shortcomings of in-context learning and supervised fine-tuning in improving LLMs on our high-level subtasks (RQ 5, RQ 6). This aims to underscore the **intrinsic limitations** of SOTA LLMs in achieving deep understanding.

**Experimented Models**  We use commercial LLMs, including GPT-3.5 (`gpt-3.5-turbo-1106`), GPT-4/4v (`gpt-4-turbo-2024-04-09`) and GPT-4o (`gpt-4o-2024-05-13`); and open-source LLMs, including Llama-3 (`Llama-3-8B-Instruct`) (MetaAI, 2024) and Mistral (`Mistral-7B-Instruct-v0.2`) (Jiang et al., 2023), InternVL-Chat-V1-5  (Chen et al., 2023; 2024)and LLaVA-NeXT-34B (Liu et al., 2023a;b). We use the default inference configurations of the LLMs. Considering the randomness, we run each experiment 3 times and compute the average and standard derivation.

## 5   VALIDATION ON LOW-LEVEL SUBTASKS

To illustrate the stochastic parrot phenomenon with PHYSICO, a necessary condition is to ensure the LLMs can perform well on the low-level understanding subtasks, *i.e.*, whether LLMs exhibit strong skills of *recalling* and *describing* the definitions, core properties and representative examples of the physical concepts in our tasks. That is:

**RQ 1:** *Can LLMs perform well on low-level subtasks, i.e., understanding the definitions of physical concepts in natural language?*

To answer RQ 1, we evaluate the LLMs' abilities to comprehend the definitions of these concepts and generate their descriptions and examples in natural language, as defined in Section 3.2.1.

### 5.1   CONCEPT SELECTION SUBTASK

**Settings**  We provide the standard definition of a concept based on Wikipedia with its synonyms masked; then ask the LLMs to identify the concept, under the same four-choice setting throughout the experiments. We evaluate the representative text-only LLMs and compute the accuracy.

**Results**  Table 1 shows that the closed-source GPT (both text-based and visual-based) models perform near perfect on recognition of our physical concepts from standard text-based definitions and from the real-life images. Moreover, we observed that open-source models make more mistakes compared with the closed-source models due to the smaller model size. For the

| (a) | **Mistral** | **Llama-3** | **GPT-3.5** | **GPT-4** |
|-----|-------------|-------------|-------------|-----------|
|     | $81.0_{\pm 1.3}$ | $88.5_{\pm 0.7}$ | $97.3_{\pm 0.3}$ | $95.0_{\pm 0.9}$ |

| (b) | **InternVL** | **LLaVA** | **GPT-4v** | **GPT-4o** |
|-----|--------------|-----------|------------|------------|
|     | $66.3_{\pm 7.7}$ | $66.7_{\pm 5.8}$ | $93.7_{\pm 0.9}$ | $93.7_{\pm 0.5}$ |

Table 1: Accuracy on the text-based (a) and visual-based (b) concept selection subtasks.

text-based models, both Mistral and Llama-3 are not as good as the closed-source models. Surprisingly, both InternVL and LLaVA are much worse than the open-source GPT models. One possible reason to this discrepancy is that our text-based concepts are from Wikipedia which is usually used as a part of the training data for open-source LLMs. In contrast, some of our selected images for those concepts may not be included in the training data of both InternVL and LLaVA which thereby can not memorize those visual instances.

### 5.2   CONCEPT GENERATION SUBTASK

**Settings**  This subtask evaluates the descriptions LLMs generate for a concept. The evaluation of a text generation task is in general difficult. Moreover, in our scenario each concept have many different ground-truth examples in its description, thus existing automatic metrics such as BLEU (Papineni et al., 2002) and METEOR (Banerjee & Lavie, 2005) are not capable of accurately measuring the quality. Therefore, we rely on mainly human evaluation for this subtask. We also propose an automatic metric via a self-play game for completeness in Appendix B.3.

**Human evaluation metric** We ask the annotators to evaluate the quality of the generated descriptions. The evaluation uses binary scores: each description receives a score of 0 if it consists of any factual error on the concept itself or any unfaithful examples,[4] and a score of 1 otherwise.

**Results** The results of automatic and human evaluations are shown in Table 2. According to human evaluation, there are no factual errors in the generated descriptions except for Mistral, confirming that our selected concepts rely on basic and widely accepted knowledge. Thought accurate, the open-source LLMs

| Mistral | Llama-3 | GPT-3.5 | GPT-4 |
|---------|---------|---------|-------|
| 92.6 | 100 | 100 | 100 |

Table 2: Human evaluations on the concept generation subtask.

sometimes include correct but uncommon facts, *e.g.,* listing single-slit diffraction as an example of *Wave Interference*. The additional results of the self-play test in Appendix B.3 further justify that all LLMs can accurately recognize the physical concepts from the descriptions they wrote by themselves. Combining the conclusions, it shows the LLMs can generate correct and sufficient information.

**Remark** We also ask the annotators of our PHYSICO-CORE to evaluate whether the core properties they annotated are covered by the LLMs' generated descriptions. This corresponds to measuring the recall of the generated descriptions on core properties/examples of concepts from PHYSICO-CORE. The recall rates for GPT-3.5 and GPT-4 are *85.0* and *90.0*, respectively. Of course, there are some exceptional examples from PHYSICO-CORE missed in the descriptions. One example is that the LLMs fails to draw the connection between *movable pulley* and the *Lever* concept. Moreover, by manually checking these missed properties and examples, we found that most of them can be recalled if we query the LLMs in a second turn by prompting "Any more core properties or examples?". This confirms that the LLMs are *aware of* and are *able to recall* the core properties of concepts covered by the PHYSICO-CORE, though some of them may not have the top conditional probabilities of generation.

**Conclusion** LLMs understand the concepts covered by PHYSICO in natural language format. Notably, we find that the properties and examples annotated in PHYSICO-CORE are *within the LLMs' knowledge* and are *highly likely to pop up* when the corresponding physical concepts are queried.

# 6 EXPERIMENTS ON HIGH-LEVEL SUBTASKS

This section answers the research questions regarding our high-level understanding subtasks.

**RQ 2:** *Can Humans understand the abstract representations?*

First of all, we investigate the performance of humans who possess the knowledge required by our PHYSICO. For each instance in our PHYSICO, we asked three independent annotators who were *not* involved in our task design to perform the same classification task presented to the LLMs. The results indicate that our tasks are largely solvable to people with a college-level education. Specifically, on the PHYSICO-CORE tasks, humans achieved an accuracy rate higher than 90%. The PHYSICO-ASSOCIATIVE tasks present greater challenges and subjectivity because the annotations are personalized based on the annotators' individual perspectives and experiences. Despite these challenges, humans can still achieve a notable average accuracy of 77.8% in solving these tasks.

We conducted a detailed investigation into human performance on a subset of PHYSICO-ASSOCIATIVE. Participants were asked to annotate instances where they believed none of the four candidate answers adequately explained the inputs. The results revealed a 10.4% rate of disagreement. On these disagreed-upon examples, human accuracy was 33.3%, explaining a major factor for the human performance decline.

**Conclusion** Our study demonstrates that humans can perform the PHYSICO tasks quite well.

**RQ 3:** *Can LLMs understand concepts in the abstract representations of the matrix format?*

A straightforward solution for our PHYSICO is to represent the grid-formatted examples as matrices. By representing the matrices with a token sequence, they can be integrated into an instruction prompt

---

[4]For example, if the LLMs generated a wrong year in the description, it is not counted as incorrect physical knowledge.

for text-based LLMs, following existing prompting methods for ARC tasks (Acquaviva et al., 2022; Xu et al., 2023; Wang et al., 2023; 2024). We use the prompt shown in Figure 7 to query the answers from the evaluated LLMs.

**Results**  The top (text-only) section of Table 3 presents the results. All the LLMs perform poorly on the three sets of our PHYSICO. Notably, GPT-3.5, Mistral, and Llama-3 failed to show significant improvement over random performance. Even for the remarkable GPT-4, GPT-4o and GPT-4v, their performance is not descent and particularly there is a huge performance gap between them and humans. In addition, as our PHYSICO is essentially an inductive reasoning task from grid-represented examples, we also tested the recent o1-preview model with strong reasoning capability. Because o1-preview is very slow and especially has limited quota, we compare it on a subset of 50 instances, on which its accuracy is only 42.0, showing no improvement over GPT-4 (with accuracy of 44.0) and GPT-4o (with accuracy of 56.0). The detailed performance decomposition of GPT-4, GPT-4o and o1-preview can be found in Appendix D.

| Models | Dev CORE-Dev | Test CORE-Test | Test ASSOC. |
|---|---|---|---|
| Random | 25.0 | 25.0 | 25.0 |
| **text-only** | | | |
| GPT-3.5 | $24.4_{\pm 0.8}$ | $26.5_{\pm 2.5}$ | $30.0_{\pm 2.5}$ |
| GPT-4 | $28.2_{\pm 2.3}$ | $41.3_{\pm 1.3}$ | $38.3_{\pm 1.2}$ |
| GPT-4o | $31.3_{\pm 2.9}$ | $34.0_{\pm 2.9}$ | $35.5_{\pm 2.5}$ |
| o1-pre* | – | 42.0* | – |
| Mistral | $26.0_{\pm 1.4}$ | $21.5_{\pm 0.3}$ | $23.2_{\pm 0.4}$ |
| Llama-3 | $27.3_{\pm 0.6}$ | $23.5_{\pm 2.5}$ | $21.7_{\pm 2.0}$ |
| **multi-modal** | | | |
| GPT-4v | $28.7_{\pm 2.4}$ | $34.2_{\pm 1.6}$ | $32.0_{\pm 1.5}$ |
| GPT-4o | $\mathbf{45.2_{\pm 2.3}}$ | $\mathbf{52.3_{\pm 0.8}}$ | $36.5_{\pm 0.4}$ |
| +CoT | $43.5_{\pm 0.8}$ | $46.0_{\pm 2.5}$ | $\mathbf{39.5_{\pm 1.1}}$ |
| InternVL | $26.9_{\pm 4.1}$ | $26.3_{\pm 1.6}$ | $24.8_{\pm 1.3}$ |
| LLaVA | $28.5_{\pm 1.5}$ | $26.2_{\pm 1.1}$ | $24.7_{\pm 3.2}$ |
| Humans | – | $92.0_{\pm 4.3}$ | $77.8_{\pm 6.3}$ |

Table 3: Performance of different text-only and multi-modal LLMs on our tasks. InternVL denotes InternVL-Chat-V1-5 and LLaVA denotes LLaVA-NeXT-34B. *The o1-preview model is evaluated on a partial subset, where GPT-4 (text) and GPT-4o perform *44.0* and *56.0*.

**Conclusion**  Comparing the human performance in RQ 2 to the best-performing LLMs reveals a huge gap. While these tasks are simple or trivial for humans, LLMs face substantial challenges, indicating a lack of deep understanding.

When comparing LLMs' performance on low-level natural language tasks in RQ 1 to high-level abstract pattern understanding tasks, we observe significant declines. This highlights the presence of the *stochastic parrot* phenomenon in LLMs. Our dataset also *quantifies the severity of this phenomenon*. For example, while GPT-3.5 performs on par with GPT-4 on the low-level tasks, it nearly drops to random guessing on our high-level tasks, revealing its tendency to act as a stochastic parrot with the physical concepts in our dataset.

**RQ 4: *Can multimodal LLMs perform well on our tasks with visual input representations?***

Next, we explore whether multi-modal LLMs can effectively solve our tasks when the input examples are presented as visual images rather than matrices like in RQ 3. We use the prompt in Figure 8 to query the answers from evaluated LLMs.

**Results**  The bottom (multi-modal) section of Table 3 shows the results. Consistent with the observations in RQ 3, a significant gap between the performance of LLMs and humans exists.

Notably, the recently introduced GPT-4o outperforms all other LLMs on PHYSICO-CORE by 10% with visual inputs but lags behind GPT-4 on matrix inputs. This discrepancy may be due to GPT-4o's training on images that directly illustrate physical concepts, making it more adept at solving problems like in Figure 1. However, this advantage does not extend to the more abstract problems in PHYSICO-ASSOCIATIVE that require further knowledge application skills, highlighting the LLMs' lack of deep understanding even with multi-modal training.

Finally, given that LLMs can generate high-quality descriptions of the concepts (see RQ 1), we adopt a chain-of-thought (Wei et al., 2022) approach. It first asks the LLMs to describe each choice and then makes predictions. The results in Table 3 (+CoT) show limited improvement or performance drop, further confirming the presence of the stochastic parrot phenomenon.

**RQ 5:** *Is* PHYSICO *challenging mainly due to LLMs' unfamiliarity with grid representations?*
One might argue that the challenges of PHYSICO might be due to the uncommon nature of the task format (especially the matrix-format inputs) encountered during LLM training, rather than a lack of deep understanding. We disprove this hypothesis from two perspectives:

*(1) We show that **GPT-4o is actually familiar with the grid representations to some extent**.* Specifically, we conducted a human study to examine GPT-4o's fundamental visual comprehension skills (Girshick et al., 2014; Long et al., 2015; He et al., 2017), including recognizing objects from the grids, describing their colors and shapes, and identifying which objects have their color, shape, or position changed from input to output. These tasks correspond to the fundamental computer vision tasks of segmentation and object detection.

We sampled 60 examples of grid pairs from our dataset and had 3 annotators determine if GPT-4o provides correct answers. For each object, the answer is counted as correct only if the shape, color, and positions are all answered correctly. Our results show an accuracy of 86.7%, which is significantly better compared to the accuracy on our high-level tasks. This confirms that GPT-4o is indeed familiar with the grid inputs but still cannot handle our PHYSICO tasks effectively.

*(2) We show that **making the LLMs more familiar with the grid representations does not lead to significant improvement**.* Specifically, we conduct the following experiments with text-only LLMs:

- *ICL on other concepts.* Compare the performance of zero-shot GPT-4 with GPT-4 using in-context learning (ICL) on few-shot examples from concepts other than the assessed one.
- *FT on synthetic matrix data.* Compare the open-source LLMs before and after fine-tuning on a large amount of matrix-input data (Appendix E)
- *FT on the ARC task.* Compare the open-source LLMs before and after fine-tuning on the original ARC (Chollet, 2019) task, which ensures that all inputs from the PHYSICO-ASSOCIATIVE examples have been seen during model training.

| Models | CORE | ASSOC. |
|---|---|---|
| GPT-4 | $41.3_{\pm1.3}$ | $39.0_{\pm0.6}$ |
| w/ ICL-3-shot | $39.5_{\pm1.6}$ | $36.2_{\pm1.7}$ |
| w/ ICL-9-shot | $32.8_{\pm1.0}$ | $39.0_{\pm1.6}$ |
| Mistral | $21.5_{\pm0.3}$ | $23.2_{\pm0.4}$ |
| w/ FT on syn-tasks | $20.9_{\pm0.7}$ | $22.5_{\pm0.5}$ |
| w/ FT on ARC | $20.9_{\pm0.8}$ | $25.5_{\pm0.9}$ |
| Llama-3 | $23.5_{\pm2.5}$ | $21.7_{\pm2.0}$ |
| w/ FT on syn-tasks | $23.0_{\pm1.1}$ | $23.2_{\pm2.7}$ |
| w/ FT on ARC | $22.2_{\pm1.6}$ | $22.4_{\pm1.2}$ |

Table 4: Performance of LLMs with in-context learning or fine-tuning on grid-format data.

Despite that both the ICL and SFT approaches make LLMs more familiar with matrix-format inputs, neither approach significantly improves the results as shown in Table 4.

**Conclusion** GPT-4o is somehow familiar with the grid format and further enhancing the familiarity of grid format for LLMs is not the key to addressing our challenges.

**RQ 6:** *How much can LLMs benefit from training on labeled data?*

Many tasks that challenge LLMs can see significant performance boosts through ICL or SFT on labeled data (Hessel et al., 2023; Yu et al., 2023; Berglund et al., 2023). When such improvements are observed, it suggests that LLMs inherently possess the necessary skills to excel in their tasks, needing only minimal training effort.

In this study, we demonstrate that this is not the case for our tasks, where light-weight training on labeled data does not improve LLM performance for our tasks. Given the current lack of large-scale training data for our purpose, we conduct an extreme case study: models learn from the same concepts in PHYSICO-CORE and are tested on the same concepts in PHYSICO-ASSOCIATIVE. To this end, we select the instances that consists of at least two choices that exist in the PHYSICO-CORE, leaving 80 examples. We conduct the following experiments on this subset to answer RQ 6:

- *ICL on the same concepts.* Compare the zero-shot GPT-4/4o and GPT-4/4o with ICL[5] on examples for the same concepts from PHYSICO-CORE. Specifically, for each instance, we sample 9 examples from PHYSICO-CORE with their labels among the choices of the instance.

---

[5]For GPT-4o, we implement ICL with multi-turn dialogues. Each shot in the demonstration is provided in one turn which asks the GPT-4o to explain the image.

| GPT-4 | $42.9_{\pm2.4}$ | GPT-4o | $40.4_{\pm2.1}$ | Llama-3 | $22.1_{\pm2.8}$ |
| + ICL on CORE | $40.0_{\pm1.0}$ | + ICL on CORE | $37.1_{\pm2.6}$ | + SFT on CORE | $20.9_{\pm2.7}$ |

Table 5: Accuracy on the subset of ASSOCIATIVE subtask that has overlapped concepts with CORE.

- *SFT on the* CORE *set.* Compare the open-source LLMs before and after fine-tuning on labeled data from PHYSICO-CORE.

**Results**   Table 5 shows that ICL and SFT on the labeled examples of the same concepts lead to a consistent, though not severe, drop in performance. The results suggest that the models have become overfitted to the "clean" examples from the PHYSICO-CORE. They appear to have learned superficial correlations from the demonstrations that do not generalize well, providing further evidence of the stochastic parrot phenomenon. The difficulty of generalization *within the same concepts* indicates the challenges of our tasks to the supervised fine-tuning paradigm.

**Conclusion**   Together with the results for RQ 5 and RQ 6, it suggests that the low performance of LLMs is not likely to be improved from prompting techniques alone. There exists intrinsic inefficiency in the pre-training of LLMs, which results in the lack of necessary skills for deep understanding.

## 7   RELATED WORK

**Stochastic Parrots on LLMs**   The pioneer study by (Bender & Koller, 2020) questioned the understanding ability of large models; and Bender et al. (2021) first introduced the terminology of stochastic parrot. The concept of stochastic parrot has received great attention, leading to a surge of studies on this topic. According to Google Scholar, the term "stochastic parrot" appears in the titles of dozens of papers from diverse research fields (Borji, 2023; Li, 2023; Duan et al., 2024; Henrique et al., 2023). However, although the concept of stochastic parrots in LLMs is widely accepted and recognized, to the best of our knowledge, there is a lack of quantitative experiments to precisely verify this viewpoint. This gap directly motivates our work.

**Abstract Reasoning Challenge**   Abstract reasoning challenge (ARC) aims to examine the inductive reasoning ability in a few-shot scenario (Chollet, 2019) and it has been used as a remarkable testbed to measure the intelligence of LLMs. Recently, many research efforts have been made on improving the performance of LLMs on ARC benchmark (Tan & Motani, 2023; Wang et al., 2023; Xu et al., 2023; Mirchandani et al., 2023; Wang et al., 2024; Huang et al., 2024). We draw inspiration from ARC by utilizing input-output grids as abstract representations in our task design. However, our task is significantly different from the ARC-style work — our high-level understanding task focuses on comprehending the transformation rules from inputs to outputs and relating them to physical concepts, and is designed to assess the stochastic parrot phenomenon.

**Challenging Tasks towards LLMs' Understanding**   Extensive recent efforts have been made on designing tasks that challenge the understanding abilities of LLMs (Chakrabarty et al., 2022; Tong et al., 2024; Shapira et al., 2023; Hessel et al., 2023; Donadel et al., 2024; Li et al., 2024). For example, Hessel et al. (2023) proposed a humor understanding task, revealing a large performance gap between LLMs and humans. As a by-product, our PHYSICO challenges the understanding capabilities of LLMs, relating it to the above studies. However, we make primary contribution to provide an quantitative experiment to verify stochastic parrots in LLMs via controllably paired low-level and high-level tasks.

## 8   CONCLUSION

We introduce PHYSICO, a novel task to assess machines' understanding of physical concepts at different levels. Our experiments reveal that: 1) LLMs lag significantly behind humans on PHYSICO, indicating a lack of deep understanding of the covered concepts; 2) LLMs exhibit the stochastic parrot phenomenon, as they excel at low-level remembering tasks but struggle with high-level understanding tasks; 3) LLMs' poor performance stems from its intrinsic deficiencies, as neither in-context learning nor fine-tuning improves their results.

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

## A  DETAILS OF THE INCLUDED CONCEPTS IN OUR PHYSICO

**Concepts in PHYSICO-CORE**   The concepts in PHYSICO-CORE are basic physical concepts that we manually design problems for. The development set covers 25 concepts and the test set covers 27 concepts as follows:

| | | | |
|---|---|---|---|
| atmospheric pressure | 12 | energe conservation | 10 |
| elastic force | 10 | friction | 9 |
| photoelectric effect | 8 | heat conduction | 8 |
| doppler effect | 8 | electromagnetic wave | 8 |
| melting | 8 | vaporization | 8 |
| fluid pressure | 8 | thermal expansion and contraction | 8 |
| Brownian motion | 8 | splashing | 8 |
| oscillation | 8 | relativity | 8 |
| lighting | 8 | lifting | 8 |
| force composition | 8 | pulley | 8 |
| inclined plane | 8 | Bernoulli effect | 7 |
| fictitious force | 6 | siphon | 6 |
| resonance | 4 | | |

Table 6: Concepts and their corresponding number of instances in PHYSICO-CORE-Dev.

| | | | |
|---|---|---|---|
| reference frame | 12 | gravity | 10 |
| reflection | 10 | refraction | 10 |
| light imaging | 10 | communicating vessels | 10 |
| cut | 10 | laser | 10 |
| surface tension | 10 | move | 10 |
| buoyancy | 10 | acceleration | 10 |
| inertia | 10 | electricity | 10 |
| repulsive force | 8 | wave | 8 |
| lever | 6 | optical filters | 6 |
| compression | 4 | diffuse reflection of light | 4 |
| wave interference | 4 | diffusion | 4 |
| vortex | 4 | expansion | 4 |
| nuclear fission | 2 | nuclear fusion | 2 |
| diffraction of waves | 2 | | |

Table 7: Concepts and their corresponding number of instances in PHYSICO-CORE-Test.

**All Concepts in PHYSICO**   The following table summarized all the concepts from both PHYSICO-CORE and PHYSICO-ASSOCIATIVE:

## B  DETAILS OF ANALYSIS METHODS IN RQ 1

### B.1  MASKING OF TEXTUAL DESCRIPTIONS

This experiment follows the setting in the "Physical Concept Selection Subtask" in section 3.2.1. The definitions of the corresponding phenomena were extracted from Wikipedia as well as generated by GPT-3.5 and GPT-4. To maintain consistency, the terms representing concepts were masked as [PHENOMENON] while relevant terms are masked as [MASK]. For instance, "interference" which corresponds to the phenomenon "wave interference" was masked as [PHENOMENON]. In contrast, "Newton's first law of motion" which corresponds to the phenomenon "inertia" was masked as [MASK].

An example of the masked description can be found in Figure 6.

| | | | |
|---|---|---|---|
| laser | 30 | mirror | 30 |
| wave | 21 | reference frame | 20 |
| gravity | 19 | move | 18 |
| reflection | 15 | zoom in | 15 |
| compression | 14 | magnet | 14 |
| expansion | 13 | explosion | 11 |
| refraction | 10 | light imaging | 10 |
| communicating vessels | 10 | cut | 10 |
| surface tension | 10 | buoyancy | 10 |
| acceleration | 10 | inertia | 10 |
| electricity | 10 | rotation | 10 |
| repulsive force | 8 | diffusion | 8 |
| optical filters | 7 | water ripples | 7 |
| long exposure | 7 | lever | 6 |
| wave interference | 5 | vortex | 5 |
| wetting | 5 | diffuse reflection of light | 4 |
| nuclear fission | 3 | nuclear fusion | 3 |
| zoom out | 3 | diffraction of waves | 2 |
| projection | 2 | polarization of light | 1 |
| chemical bond | 1 | squeeze | 1 |
| lumination | 1 | vacuum | 1 |

Table 8: Concepts and their corresponding number of instances in PHYSICO-ASSOCIATIVE.

### B.2 PROMPTS USED FOR DESCRIPTION GENERATION AND CLASSIFICATION

Figure 4 and 5 include the prompts used for generation and classification respectively.

```
[SYSTEM]
You are an expert in physics. You task is to provide a comprehensive definition of a given physical
↪ concept or phenomenon, with the key properties or key examples of the concept included.

[USER]
Please provide me with the definition of the physical concept "{{ CONCEPT }}", with the key properties
↪ or key examples included.
```

Figure 4: The prompt template used for generating descriptions of physical concepts (denoted as the variable CONCEPT) in RQ 1.

```
[SYSTEM]
You will be playing a game:
You are given a definition of a physical phenonmenon, where the names of the phenonmenon are masked.
Your task is to guess which phenonmenon the definiton refers to.
Please select the most close answer from the provided options.

[USER]
Here is a definition of a physical phenonmenon, where the names of the phenonmenon are masked:

[Definition]

{{ MASKED DESCRIPTION }}

Please guess which phenonmenon the definiton refers to. You should choose your answer from the
↪ following options: {{ CANDIDATE ANSWERS }}

Your response should end with your choice of answer.
```

Figure 5: The prompt template used for guessing the referred physical concept from four candidates (denoted as the variable CANDIDATE ANSWERS) from the natural language descriptions (denoted as the variable MASKED DESCRIPTION) in RQ 1.

```
[PHENOMENON] is a fundamental concept in physics that describes the resistance of any physical object
↪ to a change in its state of motion. This concept is a central part of [MASK], often referred to as
↪ the law of [PHENOMENON]. According to this law, an object at rest will stay at rest, and an object
↪ in motion will continue to move at a constant velocity, unless acted upon by a net external force.
↪ Here are the key properties and examples of [PHENOMENON]:

### Key Properties:
1. **Dependence on Mass**: The [PHENOMENON] of an object is directly proportional to its mass. The
↪ greater the mass of an object, the greater its [PHENOMENON], and hence, the more force it requires
↪ to change its state of motion.

2. **Resistance to Acceleration**: [PHENOMENON] is essentially the resistance of an object to any
↪ change in its velocity, which includes changes in the speed or direction of the object's motion.

3. **Universal Applicability**: [PHENOMENON] applies to all objects with mass, whether they are
↪ microscopic or astronomical in scale.

4. **Independence from External Factors**: The [PHENOMENON] of an object is inherent and does not
↪ depend on external conditions such as the environment, temperature, or pressure.

### Key Examples:
1. **A Parked Car**: A parked car will not move unless a force is applied to it. Once moving, it will
↪ continue to move at a constant speed in a straight line unless forces like friction or brakes are
↪ applied to change its state.

2. **Astronauts and Objects in Space**: In the vacuum of space, where there is little to no external
↪ force, an astronaut or any other object will continue moving in the same direction and at the same
↪ speed until acted upon by another force. This is an example of [PHENOMENON] in a microgravity
↪ environment.

3. **Seatbelts in Vehicles**: When a car suddenly stops, the passengers inside tend to lurch forward.
↪ This is due to the [PHENOMENON] of their bodies; their bodies were in motion and tend to remain in
↪ motion despite the car stopping. Seatbelts provide the necessary force to counteract this [
↪ PHENOMENON] and keep the passengers safe.

4. **Tablecloth Trick**: A classic example demonstrating [PHENOMENON] is the tablecloth trick, where a
↪ quick pull of the tablecloth can leave dishes undisturbed on a table. The [PHENOMENON] of the dishes
↪  (their tendency to resist changes in motion) allows them to remain relatively still while the
↪ tablecloth is quickly pulled from under them.

Understanding [PHENOMENON] is crucial for analyzing the motion of objects in various physical contexts,
↪  from everyday life to complex scientific scenarios. It is a cornerstone in the study of dynamics
↪ and plays a critical role in engineering, automotive safety, aerospace technology, and many other
↪ fields.
```

Figure 6: An example of our masked description for the concept `inertia`.

## B.3 ADDITIONAL RESULTS ON THE SELF-PLAY GAME

Automatic evaluation of a text generation task is in general difficult. Especially, in our scenario each concept have many different ground-truth examples in its description, thus existing automatic metrics such as BLEU (Papineni et al., 2002) and METEOR (Banerjee & Lavie, 2005) are not capable of accurately measuring the quality. Therefore, we propose an alternative automatic metric via a self-play game for this subtask:

For each generated description of a concept, we mask the synonyms of the concept in it as in the previous selection subtask, and ask the same LLM to identify the concept being described from four options. This metric evaluates the quality of LLMs' generated concept descriptions in an objective manner.

**Results** The results of automatic evaluation via self-play are shown in Table 9 together with the human evaluation results. In the self-play test, all LLMs can accurately recognize the physical concepts from the descriptions they wrote by themselves. Combined with the conclusion from human evaluation, it shows the LLMs can generate correct and sufficient information.

|  | **Mistral** | **Llama-3** | **GPT-3.5** | **GPT-4** |
|---|---|---|---|---|
| Human | 92.6 | 100 | 100 | 100 |
| SP | $89.2_{\pm1.6}$ | $91.9_{\pm0.6}$ | $96.0_{\pm0.4}$ | $99.8_{\pm0.2}$ |

Table 9: Evaluations on the concept generation subtask, with metrics of Self-Play success and Human evaluation.

```
[SYSTEM]
You will be playing a game:
You are given several examples. Each example consists of an ``input grid'' and an ``output grid'' of
↪ numbers from 0-9, where each number corresponds to a color.
Your task if try to find the common patterns from the examples and abstract the meanings of the
↪ patterns in the physical or mathematics world.
Based on the recognized meaning, please select the most close description of the common pattern from
↪ the provided options.

[USER]
Lets play a game where you are transforming an input grid of numbers into an output grid of numbers.

The numbers represent different colors:
0 = black
1 = blue
2 = red
3 = green
4 = yellow
5 = gray
6 = magenta
7 = orange
8 = cyan
9 = brown

Here are examples of input grids and its corresponding output grids:

Example input grid:
{{ INPUT GRID1 }}

Example output grid:
{{ OUTPUT GRID1 }}

Example input grid:
{{ INPUT GRID2 }}

Example output grid:
{{ OUTPUT GRID2 }}

Example input grid:
{{ INPUT GRID3 }}

Example output grid:
{{ OUTPUT GRID3 }}

Please first try to find the common patterns from the input-output pairs, then answer the following
↪ question:

What meanings in the physical or mathematics world can be abstracted from the patterns? Please choose
↪ your answer from the following options:
{{ CANDIDATE ANSWERS }}

Your response should end with your choice of answer.
```

Figure 7: The prompt template used in RQ 3. The pair of an **INPUT GRID** and an **OUTPUT GRID** consists of one example of a physical phenomenon in matrix format.

## C  DETAILS OF THE METHODS USED IN RQ 3 AND RQ 4

We use the prompt template in Figure 7 for experiments on text-only LLMs (RQ 3); and the template in Figure 8 for multi-modal LLMs (RQ 4).

## D  PERFORMANCE DECOMPOSITION IN RQ 3 AND RQ 4

Table 10 provides a performance decomposition of text-based GPT-4, text-based o1-preview and multi-modal GPT-4o on our PHYSICO-CORE-Test set. Because the rate limit of o1-preview, we conduct experiment on a subset of 50 instances. The result shows that o1-preview does not show superior results compared to the other two LLMs.

```
{{ UPLOADED IMAGE }}
[USER]
In the given image, there are two columns of matrices with elements represented by different colors.
The left column represents the inputs, and the right column represents the corresponding outputs.
For each row in the image, the output is derived from the input using the same transformation rule,
which corresponds to a real-world physical concept.

Your task is to identify the physical concept demonstrated in this image from the following options:

{{ CANDIDATE ANSWERS }}

Please select and provide the correct option that matches the transformation shown in the image.
Your response should end with your choice of answer.
```

Figure 8: The prompt template used in RQ 4. `UPLOADED IMAGE` is an image consists of three or more examples like in Figure 2.

| Concept | GPT-4 (text) | GPT-4o (visual) | o1-preview (text) |
|---|---|---|---|
| gravity | $60.0_{\pm 8.2}$ | $33.3_{\pm 4.7}$ | 50.0 |
| compression | $50.0_{\pm 20.4}$ | $50.0_{\pm 0.0}$ | 0.0 |
| diffuse reflection of light | $50.0_{\pm 0.0}$ | $33.3_{\pm 11.8}$ | 25.0 |
| lever | $0.0_{\pm 0.0}$ | $50.0_{\pm 0.0}$ | 16.7 |
| wave interference | $83.3_{\pm 11.8}$ | $100.0_{\pm 0.0}$ | 100.0 |
| spectrum of light and optical filters | $66.7_{\pm 0.0}$ | $88.9_{\pm 15.7}$ | 66.7 |
| surface tension | $43.3_{\pm 17.0}$ | $50.0_{\pm 8.2}$ | 30.0 |
| nuclear fission | $16.7_{\pm 23.6}$ | $100.0_{\pm 0.0}$ | 100.0 |
| nuclear fusion | $0.0_{\pm 0.0}$ | $100.0_{\pm 0.0}$ | 50.0 |
| communicating vessels | $3.3_{\pm 4.7}$ | $3.3_{\pm 4.7}$ | 0.0 |
| diffraction of waves | $83.3_{\pm 23.6}$ | $100.0_{\pm 0.0}$ | – |
| reflection | $86.7_{\pm 4.7}$ | $43.3_{\pm 4.7}$ | – |
| refraction | $20.0_{\pm 8.2}$ | $83.3_{\pm 4.7}$ | – |
| light imaging | $10.0_{\pm 0.0}$ | $0.0_{\pm 0.0}$ | – |
| cut | $90.0_{\pm 0.0}$ | $73.3_{\pm 4.7}$ | – |
| laser | $46.7_{\pm 12.5}$ | $53.3_{\pm 4.7}$ | – |
| move | $96.7_{\pm 4.7}$ | $86.7_{\pm 4.7}$ | – |
| buoyancy | $43.3_{\pm 12.5}$ | $100.0_{\pm 0.0}$ | – |
| acceleration | $10.0_{\pm 8.2}$ | $73.3_{\pm 12.5}$ | – |
| inertia | $80.0_{\pm 8.2}$ | $6.7_{\pm 4.7}$ | – |
| electricity | $16.7_{\pm 4.7}$ | $53.3_{\pm 9.4}$ | – |
| reference frame | $27.8_{\pm 3.9}$ | $13.9_{\pm 3.9}$ | – |
| repulsive force | $20.8_{\pm 5.9}$ | $20.8_{\pm 11.8}$ | – |
| diffusion | $8.3_{\pm 11.8}$ | $100.0_{\pm 0.0}$ | – |
| vortex | $0.0_{\pm 0.0}$ | $100.0_{\pm 0.0}$ | – |
| expansion | $50.0_{\pm 0.0}$ | $75.0_{\pm 0.0}$ | – |
| wave | $16.7_{\pm 15.6}$ | $33.3_{\pm 5.9}$ | – |

Table 10: Performance decomposition to concepts on PHYSICO-CORE-Test.

# E  CONSTRUCTION OF SYNTHETIC TRAINING DATA USED IN RQ 5

We investigate whether fine-tuning LLMs on matrix property-related questions could improve their performances on our tasks. Specifically, we generate 3000 extra input-output grid pairs calculate the size, transpose, and locations of the subgrid's corner elements for these matrices as ground truths. Furthermore, since correctly recognizing the location of the subgrid may contribute more to finish the Move and Copy tasks compared to other properties, we create additional ground truths only with the gold locations of the subgrid's corner elements.

# F  HYPERPARAMETERS OF SUPERVISED FINE-TUNING IN RQ 5 AND RQ 6

For all the fine-tuning experiments, we use LoRA (Hu et al., 2021). We fine-tune each model for 3 epochs with a batch size of 4 on a single machine with 8 A100 GPUs. The dimension of LoRA's

attention layer is set to 64, and the $\alpha$ and dropout rates are set to 16 and 0.1, respectively. The learning rate and weight decay are set to 2e-4 and 0.001, respectively. The hyperparameters are selected according to the development performance on the synthetic matrix data in Appendix E.

