# OpenReview forum: "The Stochastic Parrot on LLM’s Shoulder:  A Summative Assessment of Physical Concept Understanding"
_ICLR.cc/2025/Conference — ICLR 2025 Conference Withdrawn Submission_

### Official Review · Reviewer_kUQZ · 2024-10-16

**Soundness:** 2
**Presentation:** 2
**Contribution:** 3
**Rating:** 3
**Confidence:** 4

**Summary:**

This paper introduces PHYSICO, a novel task designed to assess LLMs' understanding of physical concepts at different levels. The authors conduct experiments comparing LLM performance on low-level tasks (recalling definitions) versus high-level tasks (interpreting abstract grid representations). They find that while LLMs excel at low-level tasks, they struggle significantly with high-level understanding compared to humans. The paper argues this demonstrates the "stochastic parrot" phenomenon, where LLMs can recite information without true comprehension. The authors also show that techniques like in-context learning and fine-tuning fail to substantially improve performance, suggesting fundamental limitations in LLMs' current capabilities for deep conceptual understanding.

**Strengths:**

The paper tries to tackle a well-known set of problems of LLM that are ultimately called "stochastic parrot"-type phenomena. The experiments seem well-designed with controlled comparisons between low and high level tasks. Furthermore, most comparisons include human-based verifications, establishing a clear baseline. Ultimately, the PHYSICO benchmark may be valuable for future research in this area.

**Weaknesses:**

The writing has significant room for improvement in terms of flow, clarity, and polish. Some sections are disconnected and hard to read, making it difficult for readers to conceptualize the overall argument. Unfortunately, the paper stands on shaky theoretical ground, uncritically assuming that language models could ultimately achieve human-like understanding of language. This fundamental issue is not adequately addressed. In Section 1, the authors claim that "there are no quantitative experiments to verify the stochastic parrot phenomenon in LLMs". This is simply not true. There is a large body of literature on LLM capabilities, emerging properties, theory of mind, supposed AGI, and related topics that often employ quantitative methods to verify their claims. The authors should thoroughly review this literature and revise their paper accordingly. Furthermore, I would argue that using quantitative methods alone may not be suitable for testing something that humans have but we know current architectures like Transformers cannot achieve. These models are trained with regression in mind, and that's ultimately what they achieve. Consequently, the paper feels disconnected from related work on probing LLM knowledge and reasoning capabilities. The paper lacks sufficient justification for the use of grid representations in assessing high-level understanding. While the experiment might have some merit, without further multi-modal grounding or novel design, the conclusions are very much expected given our current understanding of language model limitations. This predictability raises questions about the overall contribution and novelty of the work. The experiments' study in the paper that involve humans are not clear: how was the task explained to them? Was it similar to the prompt given to the LLM?

**Questions:**

For questions refer to the weaknesses part.

---

### Official Review · Reviewer_HKnB · 2024-10-27

**Soundness:** 3
**Presentation:** 3
**Contribution:** 3
**Rating:** 6
**Confidence:** 4

**Summary:**

This paper mainly investigates whether large language models (LLMs) have a deep understanding of physical concepts like gravity using a summative assessment. The summative assessment consists of low-level understanding tasks (e.g. using Wikipedia definitions) and high-level understanding tasks (e.g. using abstract grid representations) that humans can achieve high accuracy. The result show that LLMs have the natural language knowledge to succeed in low-level understanding tasks, but not able to apply the knowledge to solve high-level understanding tasks in both pure-text and multimodal inputs. Additionally, the proposed benchmark PhysiCo is challenging, but not because of the LLM's unfamilarity of the grid representation and in-context learning or fine-tuning to LLM for familiarization is not helpful for solving the tasks in PhysiCo. Overall, the paper demonstrates LLMs have a large gap to humans on high-level understanding of physical concepts.

**Strengths:**

**Originality**
- The idea of using grid/matrix representation (i.e. abstraction and reasoning corpus, ARC) to the physical concept domain is novel.
- PhysiCo splits into low-level tasks and high-level tasks. It is a novel two-tier method to design benchmarks because low-level tasks and high-level tasks answer different research questions and high-level tasks depend on the knowledge presented in the low-level tasks.

**Quality**
- The research questions proposed in the paper are well-designed to answer whether large language models (LLMs) understand what they say. Specifically, the paper conducted a human study to examine GPT-4o’s fundamental visual comprehension skills with the grid representations and show that it reaches a high accuracy. Moreover, the paper shows making the LLMs more familiar with the grid representation does not lead to significant improvement. These two experiments go hand-in-hand, demonstrated that using grid/matrix representation is suitable for this study for LLM's understanding physical concepts. However, there are some problems need to be further discussed.

**Clarity**
- The paper is easy to follow and well-written. The ordering of the research questions and the explanation of results are very helpful to understand the paper. Notably, the short conclusion for each research question makes the reader quick to grasp the central ideas and contributions of the paper.

**Significance**
- The Stochastic Parrot problem is long-standing and debatable to the research community of LLMs. There are many researchers in AI hold that LLM is not a stochastic parrot and it understands the language generated. However, the result of this paper demonstrates this is not true for physical understanding tasks, which is core to human cognition. Therefore, it presents significance to large language model research and gives insights and directions of improvements for future LLMs.

**Weaknesses:**

**Imbalance and insufficiency in number of instances for physical concepts**
- There is a imbalance in number of instances between different physical concepts. For example, in PhysiCo-Core-Test, there are 12 instances of reference frame but only 2 instances of nuclear fusion. Additionally, it may not be statistical significant for LLM's understanding of nuclear fusion because there are only 2 testing instances. In order to make the benchmark more statistical significant, while diversity is important, each of the physical concept should have at least 10 (ideally 30).

**Lack of separate results table for LLM's understanding of each physical concept**
- Some physical concepts like gravity are more common, but some concepts like communicating vessels are less common. It would be interesting to see a separete results table to evaluate each physical concept. Additionally, it's recommend to compare the results with humans to see if the LMs are more struggle at rare concepts, or roughly equal on all concepts. This tests the generalization ability of language models.

**Only GPT-4o is evaluated in the human study for familiarization of grid representation**
- Despite human studies are slower and more expensive to conduct, it is better to include an open-sourced model like LLaMA in the human study. GPT-4o is known to be very strong in textual understanding, so it may not be representative for other models like those smaller ones (e.g. LLaMA series).

**Lack of automatic study for familarization of the grid representation**
- Though authors use a human study for familiarization of the grid representation, it is hardly reproducible. And, use of in-context learning / finetuning not leading to better performance alone is not able to demonstrate whether the language model familiarized with the grid representation because the language model may not be able to get familiarized through either in-context learning or fine-tuning. Therefore, in order to make the argument stronger as it is the foundation of this paper, an automatic familiarization study for GPT-4o and LLaMA3 on grid representations will strengthen the paper.

**Questions:**

**Suggestions**
- (Minor) Example prompts are included in the appendix, but it's recommend to include an example of INPUT GRID and OUTPUT GRID, to make the examples more illustrative.

---

### Official Review · Reviewer_JKoy · 2024-11-03

**Soundness:** 3
**Presentation:** 3
**Contribution:** 2
**Rating:** 3
**Confidence:** 5

**Summary:**

This paper investigates the problem of whether large language models can understand physical concepts presented in abstract visual forms. The authors develop a dataset and a set of evaluation tools, named as Physico to evaluate LLMs' understanding of these concepts in a grid-format. Results show that contemporary LLMs fall far behind human performance on "high-level" tasks, and few-shot prompting can provide limited gains.

**Strengths:**

- The targeted domain -- physical concept understanding is intriguing and may have broader implications on studying LLMs' understanding. The paper is also written in a good flow and easy to read.

- The tasks are well-designed and I appreciate the effort the authors take on collecting these human annotated grid problems.

- Evaluations are performed in a quite comprehensive way, covering a range of LLMs and a satisfying human data.

**Weaknesses:**

- The most concerning thing from my side, I believe, is what are the scientific questions this paper is studying. There are two apparent gaps in authors' arguments made. The first gap is, from stochastic parrots to physical concept understanding; the second gap is, from physical concept understanding to abstract visual reasoning. The latter one is also kind of the problem the original ARC paper suffers: it remains questionable whether these LLM models are bad at reasoning about this core knowledge or just suffering from visual grounding or abstract visual understanding.

- I am also wondering what are the core contributions this paper is providing compared to the previous ARC/ConceptARC/L-ARC/BabyARC word. For example, it is studying a more constrained and limited domain -- physics, but claiming to have a broader research question -- understanding (stochastic parrots). Isn't the ARC paper, which proposes a larger set of "core knowledge", also serving as a good show case for LLMs' incapable of understanding?

**Questions:**

- How many human participants are recruited for the study?

- How do you see the ambiguity suffered in representing the physical concepts in abstract grid from, for example, in the middle column showed in Figure 1, why isn't the light red bar also falling?

---

### Official Review · Reviewer_bJ9r · 2024-11-04

**Soundness:** 3
**Presentation:** 1
**Contribution:** 2
**Rating:** 3
**Confidence:** 3

**Summary:**

The paper provides a new benchmark for physical concept understanding called PhysiCo. The benchmark puts to the test practical understanding ("high-level") vs. theoretical understanding ("low-level", or regurgitation of definitions) of physics concepts such as gravity. LLMs show low-level understanding but lack the high-level understanding of physical concepts in PhysiCo (i.e. LLMs lack generalization capabilities) and the authors connect these findings with the "stochastic parrot" phenomenon described by Bender & Koller (2021). The high-level tasks depict physical situations drawn in a grid, possibly requiring grid format proficiency. Authors show that fine-tuning or ICL techniques show little improvement in PhysiCo, making it a challenging evaluation benchmark.

**Strengths:**

* The paper presents a new benchmark for abstract physical reasoning in a format that is less explored than purely natural language (a grid format depicting a picture) and shows that LLMs yield poor performance even when passing these grids as visual inputs in multimodal models.
* The benchmark shown has multiple splits with different difficulty, grounded in Bloom's taxonomy.
* Experiments show that the benchmark is indeed very difficult even for frontier models.

**Weaknesses:**

* **The authors make the following central novelty claim: “(...) to our best knowledge, there are no quantitative experiments to verify the stochastic parrot phenomenon in LLMs.”. However, this has been extensively studied, albeit not using the term.** For example, ICLR 2024’s paper “The generative AI paradox” (West et al., 2024) explores the phenomenon of models being able to generate data that they cannot show understanding of. Moreover, research on quantifying models’ lack of generalization skills (e.g. Dziri et al., 2023) has been aiming to show that models are lacking the underlying reasoning skills and instead memorizing training data.
* This paper grounds itself in Bloom’s taxonomy to jointly present a very challenging task jointly with less demanding tasks for the same knowledge/skill set. Authors show that low level tasks have already been saturated, but more demanding ones for the same phenomenon have not. This is what has been implicitly happening in the field for the past few years when new benchmarks need to be proposed because of previous benchmark saturation (apart from other concerns, like data leakage). **Do we need a fully new task to expose this phenomenon, or could we have analyzed many such cases of saturation and associate them to different levels of the taxonomy?**
* Besides the aforementioned points, I believe the paper has merits as a new benchmark for physical reasoning. In this context, it would be important to thoroughly discuss existing research and benchmarks on these skills, e.g. Embodied AI benchmarks or LLM benchmarks for physical reasoning, like PIQA (Bisk et al., 2020). Why do we need a new benchmark on physical reasoning? What are they failing to capture?
* One of the experiments presented (RQ5) shows that models fail to solve the task even when training for it. This is presented in the context of arguing that the grid format is not a core reason for making the task difficult. It is interesting and good news that fine-tuning/ICL doesn't solve the task! However, I am not fully convinced that this implies that the model is now familiar with grids (not even if it showed non-zero performance in grids to begin with, like GPT-4o). There are complex training dynamics at play, and it is difficult to overcome pre-training bias (e.g., in Llama's pre-training there had to be a careful balance between short and long context).
* As a minor comment, I’d be cautious to accidentally anthropomorphize models, e.g. “[prior research, in contrast to this paper] do not demonstrate that LLMs claimed to understand those tasks”, which I'm confident was not the authors intention.


_References_

- West, Peter, et al. The Generative AI Paradox: "What It Can Create, It May Not Understand". ICLR 2024.
- Dziri, Nouha, et al. Faith and fate: Limits of transformers on compositionality. NeurIPS 2023.
- Bisk, Yonatan, et al. "Piqa: Reasoning about physical commonsense in natural language." AAAI 2020.

**Questions:**

* [from above section] Why do we need a new benchmark on physical reasoning? What are they failing to capture? Are they currently saturated?
* [from above section] Do we need a fully new task to expose this stochastic parrots phenomenon, or could we have analyzed many such cases of saturation and associate them to different levels of the taxonomy?
* I believe that using a grid format can be a confounder for performance even when fine-tuning, since this doesn't overcome the huge learning biases _against_ grids that were probably learned during pre-training. Besides this, what are other confounders that may arise? E.g. would grid size be a confounder for performance? (How small can these grids be while still showing that LLMs fail to reason over them? Does performance decrease sharply when increasing the grid size)

---

### Note · Authors · 2024-11-19

**Comment:**

Dear reviewers and ACs,

We sincerely appreciate the opportunity to submit and the time and effort of the reviewers and committee. After careful consideration, we decide to withdraw our submission and thanks for your understanding. Looking forward to participating in future ICLR conferences.

**Withdrawal Confirmation:**

I have read and agree with the venue's withdrawal policy on behalf of myself and my co-authors.